# Distribution of Copy Number Variation in SYT11 Gene and Its Association with Growth Conformation Traits in Chinese Cattle

**DOI:** 10.3390/biology11020223

**Published:** 2022-01-29

**Authors:** Haiyan Yang, Binglin Yue, Yu Yang, Jia Tang, Shuling Yang, Ao Qi, Kaixing Qu, Xianyong Lan, Chuzhao Lei, Zehui Wei, Bizhi Huang, Hong Chen

**Affiliations:** 1Key Laboratory of Animal Genetics, Breeding and Reproduction of Shaanxi Province, College of Animal Science and Technology, Northwest A&F University, Yangling, Xianyang 712100, China; yanghaiyanzsl@163.com (H.Y.); yuebinglin@nwafu.edu.cn (B.Y.); yangyukate@163.com (Y.Y.); candy951118@gmail.com (J.T.); yangshuling@nwafu.edu.cn (S.Y.); qiao1318471986@163.com (A.Q.); lanxianyong79@nwsuaf.edu.cn (X.L.); leichuzhao1118@nwsuaf.edu.cn (C.L.); 2Academy of Science and Technology, Chuxiong Normal University, Chuxiong 675000, China; kaixqu@163.com; 3Yunnan Academy of Grassland and Animal Science, Kunming 650212, China; 4College of Animal Science, Xinjiang Agricultural University, Urumqi 830052, China

**Keywords:** SYT11 gene, CNVs, association, cattle, gene expression

## Abstract

**Simple Summary:**

It is known that many different breeds of cattle are widely distributed in China. However, due to a lengthy selection of draught direction, there are obvious shortcomings in Chinese cattle, such as less meat production, slow weight gain, poor meat quality, and a lack of specialized beef cattle breeds. Animal breeding heavily benefits from molecular technologies, among which molecular genetic markers were widely used to improve the economic traits of beef cattle. Because the copy number variation (CNV) involves a longer DNA sequence or even the entire functional gene, it may have a greater impact on the phenotype. Recent studies have indicated that CNVs are widespread in the Chinese cattle genome. By investigating the effects of CNVs on gene expression and cattle traits, we aim to find those genomic variations which could significantly affect cattle traits, and which could provide a basis for genetic selection and molecular breeding of local Chinese cattle.

**Abstract:**

Currently, studies of the SYT11 gene mainly focus on neurological diseases such as schizophrenia and Parkinson’s disease. However, some studies have shown that the C2B domain of SYT11 can interact with RISC components and affect the gene regulation of miRNA, which is important for cell differentiation, proliferation, and apoptosis, and therefore has an impact on muscle growth and development in animals. The whole-genome resequencing data detected a CNV in the SYT11 gene, and this may affect cattle growth traits. In this study, CNV distribution of 672 individuals from four cattle breeds, Yunling, Pinan, Xianan, and Qinchuan, were detected by qPCR. The relationship between CNV, gene expression and growth traits was further investigated. The results showed that the proportion of multiple copy types was the largest in all cattle breeds, but there were some differences among different breeds. The normal type had higher gene expression than the abnormal copy type. The CNVs of the SYT11 gene were significantly correlated with body length, cannon circumference, chest depth, rump length, and forehead size of Yunling cattle, and was significantly correlated with the bodyweight of Xianan cattle, respectively. These data improve our understanding of the effects of CNV on cattle growth traits. Our results suggest that the CNV of SYT11 gene is a protentional molecular marker, which may be used to improve growth traits in Chinese cattle.

## 1. Introduction

Genetic variation is now widely used to improve meat yield and marble level in animal breeding [1]. A large number of single nucleotide polymorphisms (SNPs) were detected by high-throughput sequence and genome-wide association studies, and some of these are certainly associated with cattle growth traits [2]. Copy number variation (CNV) refers to large sequence gains and losses of 50 bp–5 Mp in the genome [3]. It involves insertion, deletion, copy, inversion, and translocation of the DNA sequence on a large scale [4]. Recently, studies have shown that CNVs have a potentially greater impact than SNPs [5].Because CNVs involve a large DNA fragment, this has potentially more severe effects, e.g., changing the structure and dose of the gene, resulting in different phenotypes [6]. A variety of complex traits and diseases can be affected by CNVs in cattle, such as breeding, immunity, muscle growth, and fat deposition [7]. More recently, researchers have discovered that some CNVs are related to important traits in animals. Deletion of 110 kb of the MIMT1 gene leads to miscarriage and stillbirth in cattle [8]. A large genomic region deleted from the ED1 gene leads to anhidrotic ectodermal dysplasia in cattle [9]. CNVs of the *MTHFSD* gene affect *MTHFSD* transcript level and litter size traits in Xiang pigs [10]. The *NTN1* gene has a specific expression in the longissimus muscle of pigs, and *NTN1*-CNV can affect intramuscular fat deposition via the AMPK pathway [11].

Synaptotagmin XI (SYT11) is located on chromosome 1q21-22 and is a member of the synaptotagmin gene family, consisting of 15 members [12]. Ca^2+^ triggers endocytosis through the voltage-dependent Ca^2+^ channel, and by calmodulin and calcineurin activation [13,14]. Synaptotagmin is necessary for Ca^2+^ dependent membrane transport in synaptic vesicles or membranes of neurons and contains two C2 (C2A and C2B) domains. Sets are the best-known regulators of exocytosis and endocytosis in the brain and endocrine cells [15,16]. The C2A domain of SYT11 contains substituted serine residues of one of the five aspartic acids required for Ca^2+^ binding, resulting in a weak Ca^2+^ binding affinity [17,18]. SYT11 usually ensures the accuracy of vesicle recovery by limiting the membrane invagination site in early endocytosis and inhibits microglial activation in physiological and pathological conditions by inhibiting cytokine secretion and phagocytosis [19,20]. One study showed that the C2B domain of SYT11 interacts with SND1, Ago2, and FMRP components of the RNA-induced silencing complex (RISC), suggesting that it may provide a link between gene regulation and membrane traffic of microRNAs [21].

miRNA is a key regulatory factor in a variety of biological pathways. miRNA mainly complements the 3′-end untranslated region (3′ UTR) of target mRNA, leading to degradation of target mRNA or inhibition of translation due to incomplete pairing [22]. It plays a certain role in the proliferation, differentiation, and regeneration of skeletal muscle cells and the regulation of muscle fiber type, in order to achieve the corresponding biological role [23].

The CNV region of SYT11 was recently discovered by sequencing the whole genome [24]. Based on the function of this gene, we speculated that the CNV of SYT11 could regulate muscle development and affect the phenotype of cattle. In this study, the SYT11 gene was used as a candidate gene to study the relationship between CNV and growth traits of Yunling (YL), Pi’nan (PN), and Xia’nan (XN) cattle. Among these, YL cattle are a three-way crossbred breed, its genome composed of one half Brahman, one quarter Murray Grey, and one quarter Yunnan Yellow cattle genome information [25]. The frozen sperm of Piedmont cattle was crossed with Nanyang cattle, then fixed and selected to obtain PN cattle [26]. In order to improve the quality of Nanyang cattle, Charolais beef blood was introduced, and a new breed of XN was cultivated by crossbreeding, reciprocal breeding, and stabilization [27]. QC cattle are one of the five local cattle breeds in China. In this study, we observed different CNV types of SYT11 and their differential effects on growth traits according to cattle hybrids and localities. These data will help to explore the use of this CNV as a molecular marker for Chinese cattle breeding.

## 2. Materials and Methods

The experimental animals and procedures of this research were approved by the Faculty Policy and Welfare Committee of Northwest A&F University. Furthermore, the feed and treatment of experimental animals adhered to the local animal welfare laws, guidelines, and policies.

### 2.1. Animal Samples and Data Collection

With the development of bioinformatics, molecular markers have gradually been applied to the selection and breeding of livestock and poultry. We randomly chose 684 adult cattle blood from four Chinese breeds: Yunling (YL, *n* = 279),Pi’nan (PN, *n* = 271), Xia’nan (XN, *n* = 87), and Qinchuan (QC, *n* = 47). YL, PN, XN, and QC cattle were reared in the provinces of Yunnan, Henan, Henan, and Shaanxi, respectively. All animals were allowed access to feed and water ad libitum under normal conditions. Growth traits of the four cattle breeds include body height, height at the hip cross, chest circumference, waist circumference, cannon circumference, chest width, chest depth, hip width, huckle bone width, Jiri length, head length, forehead size, and body weight [28,29,30].

### 2.2. Isolation of Genomic DNA and Total RNA

Genomic DNA was isolated from 684 blood samples, according to the previous method, in our laboratory [31]. The quality of genomic DNA was assayed by Nanodrop 2000, and the concentration of each DNA sample was diluted to 50 ng/µL for genomic quantitative PCR (qPCR) and was preserved in the ultra-low freezer. Skeletal muscle was obtained from adult XN cattle (*n* = 24) for genomic DNA extraction using the above method. Total RNA from these samples was isolated using TRIZOL reagent following the manufacturer’s protocol (AG, HunanChina). RNA concentration was measured by Nanodrop 2000 and reverse transcription was preformed using the *EVO M-MLV* RT Kit. After removing gDNA (AG, Hunan, China), 500 ng total RNA was used for qPCR.

### 2.3. Copy Numbers of SYT11 Gene Analysis

We used the Bovine Genome Variation Database (BGVD) to search the genomic variations in SYT11 and show the types of copy number variation of cattle around the world (Figure 1) [32]. Results of resequencing by qPCR were confirmed and validated. The primer pair was designed to target the CNV region of the *SYT11* gene by NCBI, and information on the primer is shown in Table 1. Then the PCR amplification curve and dissolution peak were observed to determine whether the primers were suitable (Figure 2A,B). We chose bovine basic transcription factor 3 (BTF3) as the internal reference gene [33] to investigate the relative copy numbers. Genomic qPCR experiments were performed in three replicate reactions using SYBR^®^ Green. A total of 10 μL reaction mixtures contained 50 ng of cDNA, 5 μL 2× ChamQ SYBR qPCR Master Mix (Vazyme, Nanjing, China), and 10 pmol of primers. Thermal cycling conditions consisted of one cycle of 30 s at 95 °C, followed by 40 cycles of 10 s at 95 °C and 30 s at 60 °C, and a melting curve of 15 s at 95 °C, 60 s at 60 °C, and 15 s at 95 °C.

### 2.4. Gene Expression Analysis

The primer sequences of the cDNA amplification are listed in Table 1, and this primer spanned an exon–exon junction from the SYT11 mRNA base sequence. The bovine glyceraldehyde-3-phosphate dehydrogenase gene (GAPDH) was used as an internal reference gene and Light Cycle 96 (Roche, Switzerland) was used for qPCR amplification. A total volume of 10 µL of reaction mixture contained 5 ng cDNA, 5 µL 2 ChamQ SYBR qPCR Master Mix (Vazyme, Nanjing, China), and 10 pmol of primers, then every sample was repeated three times. The qPCR procedure is the same as in 2.3.

### 2.5. Statistical Analysis

Association analysis has been widely used for analyzing the effect of gene polymorphism on quantitative traits in animals [34,35,36]. Target sequence and reference sequence primers were used to amplify each sample by qPCR (each group has three replications). Then, we analyzed the qPCR results by referring to previous studies in our laboratory [37]. Accordingly, the three types of copy number (gain, loss, normal) were designated as >2, <2, and =2 copies, and we calculated the distribution of different copy number variation types of each cattle breed. Subsequently, we analyzed the association between the CNV and growth traits of each type of cattle by IBM SPSS Statistics 23.0 software(SPSS, Inc., Chicago, IL, USA). Analysis of variance (ANOVA) was used to observe the effects of the CNV within the SYT11 gene on growth traits. 

However, it is also possible that our sample size was not large enough to lead to the current results, and we try to analyze all animals together and include breed effect in the model. Previously, there have been reports that gender and birth season have no obvious significance for trait variation [38]. Thus, the following model (1) was used. *Y_ijk_* is the observation of the growth traits, *μ* is population mean, *C_i_* and *B_j_* are the fixed effect of *j*th CNV type of SYT11, and *e_ijk_* is the random residual error [39].
(1)$$Y_{ijk}=\mu +C_{i}+B_{j}+e_{ijk}$$

## 3. Results

### 3.1. Distribution of the Copy Number Variation in Four Cattle Breeds

We detected the copy number of SYT11 in four cattle breeds (YL, PN, XN, and QC) containing three crossbred breeds (YL, PN, and XN) and one native breed (QC). To analyze the copy number polymorphism of the SYT11 gene in four cattle breeds, we used three ways to show the condition. As displayed in Figure 3A, the three types of SYT11 CNV frequency show different results: the frequency of gain type was the highest in YL, PN, and QC cattle, while the normal type was more than the loss type in YL cattle, but the loss type was more than the normal type in PN cattle. On the contrary, the loss type is the dominant type in XN cattle. The CNV types are classified based on log_2_2^−ΔΔCt^ as loss (<−0.5), gain (>0.5), and normal (<|0.5|). As shown in Figure 3B, the copy number gain was more dominant than loss and normal in YL, PN, and QC. In addition, it can be seen from Figure 3C that the large proportion of CNV frequency in YL and PN cattle was two, three, and four copy numbers, while one and five copy numbers occupied the largest proportion in XN and QC cattle, respectively.

### 3.2. Association Analysis between SYT11 CNVs and Growth Traits in Three Cattle Breeds

In this study, we analyzed the association of SYT11 gene copy number types with growth traits by one-way ANOVA (IBM SPSS Statistics 23.0 software). Although the gain type was the dominant trait in YL cattle, the body length and forehead size of gain type were significantly lower than the normal type, while tube circumference and hip circumference were significantly lower than loss type. This suggests that the gain type of YL cattle may have negative effects on some growth traits (Table 2). In PN cattle, the gain type was dominant, but there was no significant difference between the different copy number types by one-way ANOVA with the measured growth traits (Table 3). The bodyweight of both loss and gain type cattle was significantly higher than that of the normal type (Table 4).

In addition, we analyzed all animals together and included the breed effect in the model. We found no significant differences between the three copy number types in four cattle breeds (Table 5). However, there are significant differences in growth traits among different cattle breeds (Table 6).

### 3.3. Correlation Analysis of Copy Number Variation and mRNA Expression

Since many CNVs may have functional effects if the CNVs are in the vicinity of genes [40], in this study we focused on the effects of SYT11-CNV on the growth and development of skeletal muscle tissue, and the correlation between CNV and SYT11 mRNA level in skeletal muscle of 25 adults (XN) cattle. We found that SYT11-CNV is a loss-type, which was similar to our above analysis of copy number distribution. We further found that there was no correlation between SYT11 gene mRNA level and copy number variation. 

## 4. Discussion

CNVs are considered to be an important factor affecting livestock growth traits, in addition to single nucleotide polymorphisms (SNPs) and Indels. CNV is defined as a gain or loss of more than 50 bp nucleotides in DNA sequence between two individuals of a species [41]. The discovery of CNVs in the genome provides new points of view into genomic polymorphism. With the rapid pace of development of genomic research, CNV is related with diseases and phenotypic variation [42,43]. CNV related to susceptible genes was tested in both the diseased and normal populations, and it was found that CNV at the q21.1 position on chromosome 1 was related to neuroblastoma [44]. Insertion of the CCL3L1 gene sequence fragment is related to HIV susceptibility [45]. Lee found that CNV regions near the PLP1 gene in the genome of patients with progressive muscular dystrophy (PMD) shared deletions or repeats, and inferred that the occurrence of complex deletions and duplications could provide strong molecular genetic evidence for the etiology of the disease [46]. Moreover, CNVs in various cattle breeds have been systematically identified at the genomic level, and some loci may contribute to the phenotypic traits of cattle [3,47]. At present, a large number of studies have shown that CNVs are related to cattle production, reproduction, and health traits. For example, CNV on chromosome 5 has an impact on reproductive efficiency in Indian cattle [48]. Leptin receptor (LEPR) plays a key role in energy balance and fat development. It was found that there were significant differences in body height, body weight, and body length within different copy number types of LEPR in NY cattle [42]. 

In this study, we found that there were some differences in the dominant copy number types among different cattle breeds. YL, PN, and QC cattle were dominated by gain type, while XN cattle was dominated by loss type, and this may be related to different cattle breeds. Living environment and selection strategy of cattle may be responsible for CNV differences among these breeds [31]. Analyzing the correlation between YL cattle CNV and growth traits, we found that the body length and forehead size of the normal copy number type were higher than those of the other copy number types. The bodyweight of XN cattle with the normal copy number type was significantly lower than that with the loss and gain type. STY11-CNV was found to be located in the QTL region according to Cattle QTL Database, which is likely to affect cattle body weight (https://www.animalgenome.org/cgi-bin/QTLdb/BT/browse, accessed on 1 December 2021). Indeed, there were differences in cattle body weight among different copy number variants of XN cattle, which supported the functional roles of STY11-CNV. However, due to the small sample size of XN cattle, we are not sure whether such a result could also be obtained in large populations. Here, we also could not rule out that this may be related to regional cattle differences. YL cattle are located in the subtropical monsoon climate of Yunnan Province with a lowest monthly average temperature of above 0 °C in winter, while PN cattle and XN cattle are located in the temperate monsoon climate of Henan Province, where the lowest monthly average temperature in winter is below 0 °C. Additionally, previous evidence indicates that CNVs of dose-sensitive genes are important for normal phenotypic variation [49]. The copy number may influence gene transcription and phenotypic variation through dose-effect.

Synaptotagmin is a large class of membrane transporters thought to regulate membrane communication in both neuronal and non-neuronal cells [50,51]. SYT11 is a non-Ca^2+^ binding protein associated with schizophrenia and Parkinson’s disease, which is mainly involved in vesicle fusion, transport, and exocytosis [52]. It has been shown that constitutive knockout mice lacking SYT11 died shortly after birth, suggesting that SYT11-mediated membrane transport is necessary for mouse survival [53]. As mentioned earlier, the SYT11 C2B domain interacts with components of the RNA-induced silencing complex (RISC), and SYT11 may provide a link between gene regulation and membrane transport of microRNAs.

The purpose of this study was to determine whether SYT11 CNV also affected cattle growth traits. However, we found no significant correlation between the CNV and the mRNA expression of the SYT11 gene. Therefore, it is speculated that SYT11 CNV may affect growth traits through other regulatory mechanisms, rather than by changing mRNA levels.

## 5. Conclusions

In conclusion, this study was the first to describe the distribution of the SYT11 gene in 684 cattle of four cattle breeds in China. We found that SYT11 CNV played different roles in the three crossbred cattle breeds and speculated that CNV may be involved in breed selection. Meanwhile, our study provides preliminary results for the functional role of SYT11 CNV in larger populations and different breeds of cattle, which may provide new clues for the application of CNV as a promising molecular marker in animal breeding.

## Figures and Tables

**Figure 1 biology-11-00223-f001:**
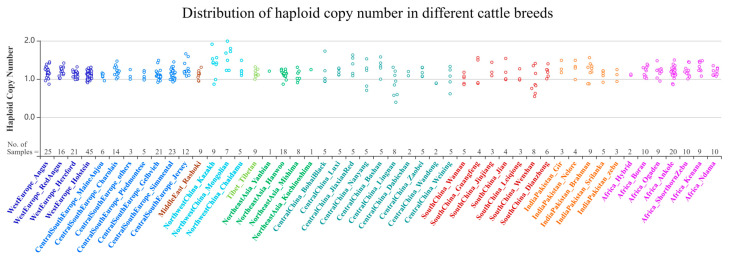
Distribution of SYT11 gene copy number in different cattle breeds.

**Figure 2 biology-11-00223-f002:**
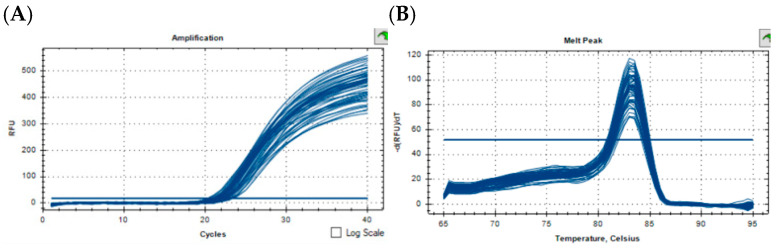
(**A**,**B**) Specificity testing of the SYT11 gene.

**Figure 3 biology-11-00223-f003:**
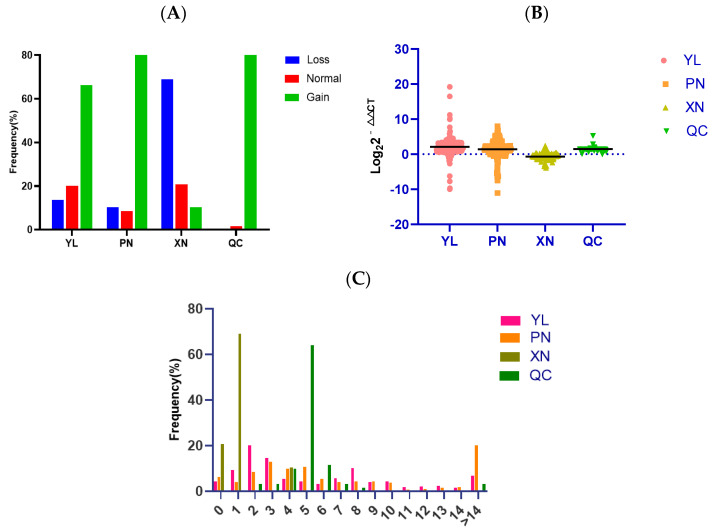
Distribution of copy number variation of the *SYT11* gene in four cattle. (**A**) The histogram shows the frequency of the *SYT11* gene in different copy number types across four cattle breeds. Copy numbers were rounded to the nearest integer. (**B**) The grouped scatterplot distribution of the *SYT11*-CNVs in four Chinese cattle breeds. YL (*n* = 276); PN (*n* = 266); XN (*n* = 80); QC (*n* = 47). (**C**) Histograms show the frequency of individuals with different copy numbers of the *SYT11* gene. Copy numbers were rounded to the nearest integer.

**Table 1 biology-11-00223-t001:** Primer pairs designed for the genes used in this study.

Detection Level	Genes	Primer Sequences (5′ to 3′)	Amplification Length	Annealing (°C)
DNA level	SYT11	F: TATTTTCCACCCCACTCTCTGC	92 bp	60
		R: TTACGTCATCTCGGAGCGGC		
	BTF3	F: AACCAGGAGAAACTCGCCAA	166 bp	60
		R: TTCGGTGAAATGCCCTCTCG		
mRNA level	*SYT11*	F: CACCTGCCGAAGATGGACATC	173 bp	60
		R: AGGTCGGTGGGGATGTCGTAG		
	GAPDH	F: TGAGGACCAGGTTGTCTCCTGCG	145 bp	60
		R: CACCACCCTGTTGCTGTAGCCA		

**Table 2 biology-11-00223-t002:** Association analysis of bovine *SYT11* gene copy number variations with growth traits in YL cattle.

Growth Traits	CNV Types (Mean ± SE)			*p*-Value
	Loss (*n* = 38)	Normal (*n* = 56)	Gain (*n* = 185)	
body height (cm)	129.88 ± 5.50	130.00 ± 6.10	127.73 ± 10.45	0.167
height at hip cross (cm)	133.55 ± 5.44	130.53 ± 20.89	131.91 ± 10.28	0.446
body length (cm)	154.11 ± 10.71 ^B^	159.68 ± 10.31 ^A^	154.27 ± 9.97 ^B^	0.010
chest circumference (cm)	198.02 ± 10.56	196.95 ± 10.93	195.55 ± 9.85	0.253
waist circumference (cm)	222.93 ± 33.083	226.53 ± 13.42	225.62 ± 20.81	0.694
cannon circumference (cm)	19.03 ± 1.28 ^A^	18.63 ± 1.58 ^A,B^	18.43 ± 1.37 ^B^	0.018
chest width (cm)	49.08 ± 4.48	49.84 ± 4.64	48.71 ± 4.17	0.328
chest depth (cm)	69.53 ± 6.37 ^a,b^	70.45 ± 5.15 ^a^	67.80 ± 5.80 ^b^	0.014
rump length (cm)	115.48 ± 12.15 ^a^	111.79 ± 10.15 ^a,b^	111.16 ± 10.78 ^b^	0.037
hip width (cm)	59.21 ± 15.52	58.34 ± 7.91	57.37 ± 4.67	0.338
hucklebone width (cm)	22.39 ± 2.48	22.29 ± 2.57	22.08 ± 2.17	0.635
head length (cm)	48.20 ± 2.80	47.92 ± 4.70	48.15 ± 3.22	0.915
forehead size (cm)	22.26 ± 1.53 ^b^	23.45 ± 4.36 ^a^	22.45 ± 1.32 ^b^	0.013
jiri length (cm)	50.16 ± 4.22	50.71 ± 3.56	49.86 ± 3.57	0.420
body weight (kg)	569.93 ± 81.57	549.46 ± 67.52	539.44 ± 110.91	0.189

Notes: Values with different superscripts (a, b) within the same row differ significantly at *p* < 0.05. Values with different superscripts (A, B) within the same row differ significantly at *p* < 0.01.

**Table 3 biology-11-00223-t003:** Association analysis of bovine *SYT11* gene copy number variations with growth traits in PN cattle.

Growth Traits	CNV Types (Mean ± SE)			*p*-Value
	Loss (*n* = 28)	Normal (*n* = 23)	Gain (*n* = 220)	
body height (cm)	123.30 ± 6.64	124.85 ± 7.52	124.61 ± 6.11	0.612
body length (cm)	144.69 ± 12.25	148.53 ± 10.27	147.19 ± 11.26	0.469
height at hip cross (cm)	129.52 ± 6.38	133 ± 6.85	131.66 ± 6.11	0.136
chest circumference (cm)	168.60 ± 12.75	174.57 ± 12.47	172.95 ± 13.14	0.234
hip width (cm)	44.73 ± 4.51	46.60 ± 4.46	45.79 ± 4.09	0.283
jiri length (cm)	46.78 ± 3.65	48.35 ± 5.45	48.11 ± 3.95	0.304

**Table 4 biology-11-00223-t004:** Association analysis of bovine *SYT11* gene copy number variations with growth traits in XN cattle.

Growth Traits	CNV Types (Mean ± SE)			*p*-Value
	Loss (*n* = 60)	Normal (*n* = 18)	Gain (*n* = 9)	
body weight (kg)	575.5 ± 59.44 ^a^	540.65 ± 59.84 ^b^	582.77 ± 49.68 ^a^	0.027
body height (cm)	136.77 ± 2.71	135.08 ± 3.65	136.77 ± 2.43	0.103
height at hip cross (cm)	138.83 ± 3.51	138.16 ± 2.91	139.11 ± 2.26	0.537
body length (cm)	161.5 ± 11.34	158.81 ± 6.98	161.11 ± 6.09	0.387
chest circumference (cm)	196.11 ± 17.25	193.1 ± 8.74	195.66 ± 8.06	0.53
waist circumference (cm)	219.33 ± 21.75	217.13 ± 17.58	222.33 ± 19.84	0.706
cannon circumference (cm)	19.5 ± 1.58	19.16 ± 1.41	19.55 ± 1.23	0.572

Notes: Values with different superscripts (a, b) within the same row differ significantly at *p* < 0.05.

**Table 5 biology-11-00223-t005:** Association analysis of bovine *SYT11* gene copy number variations with growth traits in four breeds.

Growth Traits	CNV Types (Mean ± SE)			*p*-Value
	Loss (*n* = 38)	Normal (*n* = 56)	Gain (*n* = 185)	
body height (cm)	126.45 ± 8.45	129.56 ± 6.9	131.28 ± 6.8	0.394
height at hip cross (cm)	138.17 ± 14.06	137.03 ± 8.75	138.7 ± 13.4	0.906
body length (cm)	153.34 ± 13.56	149.71 ± 15.19	141.78 ± 14.14	0.096
chest circumference (cm)	190.14 ± 13.33	190.71 ± 17.45	183.06 ± 16.54	0.814

**Table 6 biology-11-00223-t006:** Association analysis of bovine *SYT11* gene copy number variations with growth traits in four breeds.

Growth Traits	Cattle Breed (Mean ± SE)				*p*-Value
	YL (*n* = 279)	PN (*n* = 271)	XN (*n* = 87)	QC (*n* = 47)	
body height (cm)	128.47 ± 9.18 ^B^	124.53 ± 6.3 ^C^	135.61 ± 3.43 ^A^	128.04 ± 7.07 ^B^	0.000
height at hip cross (cm)	132.05 ± 11.6 ^C^	147.12 ± 11.24 ^A^	138.4 ± 2.98 ^B^	125.23 ± 7.36 ^D^	0.000
body length (cm)	154.98 ± 10.3 ^B^	131.62 ± 6.24 ^D^	159.61 ± 7.98 ^A^	136.79 ± 14.26 ^C^	0.000
chest circumference (cm)	196.24 ± 10.16 ^A^	172.75 ± 13.07 ^C^	193.99 ± 10.92 ^A^	179.57 ± 18.72 ^B^	0.000

Notes: Values with different superscripts (A–D) within the same row differ significantly at *p* < 0.01.

## Data Availability

The data that support the findings of this study are available from the corresponding author upon reasonable request.

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
