# Peer review of "Distribution of Copy Number Variation in SYT11 Gene and Its Association with Growth Conformation Traits in Chinese Cattle"

_biology, 2022, doi:10.3390/biology11020223_

Round 1
Reviewer 1 Report
The manuscript can be accepted in its present form.
Author Response
Title: Distribution of copy number variation in SYT11 gene and association with growth traits in Chinese cattle
ID: biology-1514283
Dear reviewer,
Thank you very much for your letter and suggestions.
We are so grateful for your valuable review . We have revised the article one by one in response to your suggestions. We have highlighted the modified part in yellow, and the newly added part is red. We hope that the revised manuscript could meet the publication requirements.
Thank you again for concerning our article.
Kind regards,
Hong Chen

Reviewer 2 Report
The draft is an improvement over the original one. I have made a few suggestions in the edited manuscript to help the authors improve further on the quality.

Author Response
Title: Distribution of copy number variation in SYT11 gene and association with growth traits in Chinese cattle
ID: biology-1514283
Dear reviewer,
Thank you very much for your letter and suggestions.
We are so grateful for your valuable review and careful guidance. We have revised the article one by one in response to your suggestions. We have highlighted the modified part in yellow, and the newly added part is red. We hope that the revised manuscript could meet the publication requirements.
Thank you again for concerning our article.
Kind regards,
Hong Chen
